# Targeting Uremic Toxins to Prevent Peripheral Vascular Complications in Chronic Kidney Disease

**DOI:** 10.3390/toxins12120808

**Published:** 2020-12-20

**Authors:** Chia-Lin Wu, Der-Cherng Tarng

**Affiliations:** 1Division of Nephrology, Department of Internal Medicine, Changhua Christian Hospital, Changhua 500, Taiwan; chialinwutw@gmail.com; 2Institute of Clinical Medicine, National Yang-Ming University, Taipei 112, Taiwan; 3Translational Research Laboratory, Changhua Christian Hospital, Changhua 500, Taiwan; 4Division of Nephrology, Department of Medicine, Taipei Veterans General Hospital, Taipei 112, Taiwan; 5Department and Institute of Physiology, National Yang-Ming University, Taipei 112, Taiwan; 6Center for Intelligent Drug Systems and Smart Bio-devices (IDS2B), Hsinchu 300, Taiwan; 7Department of Biological Science and Technology, College of Biological Science and Technology, National Chiao Tung University, Hsinchu 300, Taiwan

**Keywords:** AST-120, chronic kidney disease, indoxyl sulfate, peripheral vascular disease, phosphorus, uremic toxins

## Abstract

Chronic kidney disease (CKD) exhibits progressive kidney dysfunction and leads to disturbed homeostasis, including accumulation of uremic toxins, activated renin-angiotensin system, and increased oxidative stress and proinflammatory cytokines. Patients with CKD are prone to developing the peripheral vascular disease (PVD), leading to poorer outcomes than those without CKD. Cumulative evidence has showed that the synergy of uremic milieu and PVD could exaggerate vascular complications such as limb ischemia, amputation, stenosis, or thrombosis of a dialysis vascular access, and increase mortality risk. The role of uremic toxins in the pathogenesis of vascular dysfunction in CKD has been investigated. Moreover, growing evidence has shown the promising role of uremic toxins as a therapeutic target for PVD in CKD. This review focused on uremic toxins in the pathophysiology, in vitro and animal models, and current novel clinical approaches in reducing the uremic toxin to prevent peripheral vascular complications in CKD patients.

## 1. Peripheral Vascular Disease in Chronic Kidney Disease

Peripheral vascular disease (PVD), also known as peripheral artery occlusive disease, is a common but devastating disease in patients with chronic kidney disease (CKD) or end-stage renal disease (ESRD), diabetes mellitus, hypertension, and dyslipidemia, especially in elderly patients. PVD could develop early and asymptomatically but progressively lead to limb ischemia, such as intermittent claudication, pain, ulceration, and gangrene. Although it is not an immediately life-threatening disease, PVD is associated with decreased functional capacity and quality of life but higher risks for mortality and cardiovascular morbidity [1,2]. According to the Global Burden of Diseases, Injuries, and Risk Factors Study 2017, the prevalent and incident cases of PVD were at least 118 and 10.8 million and caused 515.6 thousand years lived with disability worldwide [3].

Traditional risk factors of PVD are smoking, older age, high blood pressure, diabetes, hyperlipidemia, and homocysteinemia. By using the ankle–brachial index (ABI), more recent studies have shown that the prevalence of PVD is significantly higher in patients undergoing chronic dialysis or with CKD. The Atherosclerosis Risk in Communities (ARIC) study investigated 14,280 middle-age adults with a mean follow-up of 13 years and found the incidence rates per 1000 person-years was 8.6 for participants with glomerular filtration rate between 15 and 59 mL/min/1.73 m^2^ compared with 4.7 for participants with normal kidney function [4]. The risk of developing incident PVD in patients with reduced kidney function was 56% higher than in non-CKD patients. Patients with CKD would also have nontraditional risk factors for PVD, including inflammation, oxidative stress, prothrombotic state, and insulin resistance independent of traditional risk factors [5]. A recent study even found that nearly one in ten ESRD patients eventually had amputation in their last year of life [6]. Nowadays, it is well accepted that CKD is a strong predictor or risk factor for PVD [7,8].

Diagnosis of PVD can be made by classic manifestation (intermittent claudication), noninvasive ABI, Doppler ultrasound flow study, computed tomography or magnetic resonance angiography, or invasive arterial angiography [9,10]. Among them, ABI has been recommended for screening PVD in dialysis patients by the 2005 Kidney Disease Outcomes Quality Initiative (KDOQI) guidelines. However, the Kidney Disease: Improving Global Outcomes (KDIGO) meeting report in 2011 suggested that the sensitivity and specificity of this standard diagnostic testing were not clarified for PVD in patients with CKD who may have a high probability of vascular stiffness [11]. The clinical update conference also suggested alternative diagnostic tests—toe–brachial index or pulse volume recording—though there remains a lack of large-scale studies to validate these diagnostic tests in patients with CKD [12,13].

Current treatment of PVD includes lifestyle changes, aggressive control of diabetes, blood pressure and dyslipidemia, antiplatelets, anticoagulation, cilostazol, revascularization, and amputation [10,14,15,16,17]. Although the amputation rates of patients on chronic dialysis have significantly decreased in the past two decades, the one-year mortality after lower extremity amputation remains above 40% in these patients [18]. Recent research also found that dialysis patients who underwent lower extremity amputation in the last year of life prolonged their stays in acute and subacute health care settings [6]. Therefore, there remain unmet needs for patients with CKD and ESRD who have developed, or are at high risk of, PVD [11].

So far, there is little literature addressing the management of PVD specific to CKD patients, though growing evidence suggests that uremic toxins may contribute to PVD and associate with poor clinical outcomes. In this review, we will discuss the role of uremic toxins in PVD and possible preventive or therapeutic uremic toxin-reducing strategies in patients with CKD.

## 2. Uremic Toxins and Peripheral Vascular Disease

Patients with end-stage renal disease (ESRD) have been reported to have a high prevalence of PVD, strongly suggesting the role of uremic toxins in the development and progression of the disease [19,20]. However, the actual mechanisms of how CKD contributes to the development and progression of PVD remains incompletely understood. 

Uremic toxins are harmful molecules that are mainly removed by the kidney. During the progression of CKD, uremic toxins gradually accumulate in the circulation and tissue and contribute to premature mortality and cardiovascular events [21,22,23,24]. Uremic toxins are classified into small water-soluble molecules (guanidines, purines, oxalate, phosphorous, trimethylamine N-oxide, urea, etc.), middle molecules (cystatin C, leptin, advanced glycation end products, β_2_-microglobulin, parathyroid hormone, etc.), and protein-bound molecules (indoles, hippuric acid, p-cresol, polyamines, etc.) [25]. We have reviewed the literature studying the adverse effects of uremic toxins on PVD in CKD and summarized it in Table 1.

### 2.1. Small Water-Soluble Uremic Toxins

Growing evidence has revealed possible causation between uremic toxins and the development or the progression of PVD (Table 1). Small water-soluble uremic toxins such as asymmetric dimethylarginine may have potential to induce vascular damage and are a risk factor of death in patients with PVD [44]. However, the association between these small water-soluble molecular toxins and the development of PVD is of little evidence except phosphorus. More than a decade ago, a study reported that serum phosphorus is an independent predictor of incident PVD in hemodialysis patients [26]. Ix et al. also found that patients with phosphorus levels >4 mg/dL had a 4.6-fold risk for high ABI compared with those with phosphorus levels <3 mg/dL [29]. Animal studies also demonstrated that hyperphosphatemia per se can contribute to vascular calcification [30,31]. Phosphorus promotes vascular calcification by multiple mechanisms, such as promoting osteochondrogenic differentiation and inhibiting osteoclast differentiation of vascular smooth muscle cells (VSMCs), promoting apoptosis of VSMCs, and synergizing with fibroblast growth factor 23 to activate ERK1/2 pathway in VSMCs and exaggerate calcification [27,28,31,45,46]. Moreover, trimethylamine N-oxide (TMAO) has been demonstrated to associate with atherosclerosis, reduced renal function, and mortality in patients with PVD [47,48]. Recent evidence revealed that TMAO impaired endothelium-derived hyperpolarizing factor-induced relaxation of rat femoral arteries, indicating the causative role of TMAO in the development of PVD [32].

### 2.2. Middle Molecular Uremic Toxins

Middle molecular uremic toxins might also have a deleterious effect on vessels and contribute to PVD, although evidence is sparse. A previous study reported that circulating β_2_-microglobulin levels were elevated in patients with PVD and were independently associated with the severity of the disease [33]. This group further demonstrated the inverse relationship between β_2_-microglobulin level and estimated glomerular filtration rate (eGFR) in patients with CKD [34]. In this study, β_2_-microglobulin levels were independently associated with cardiovascular events, which were a composition of traditional major cardiovascular events plus peripheral ischemia and surgical procedures for PVD in pre-dialytic CKD patients [34]. Another study demonstrated that circulating β_2_-microglobulin levels were negatively correlated with immature progenitor cell numbers in hemodialysis patients, implying that β_2_-microglobulin may be involved in the process of vascular injury [49].

### 2.3. Protein-Bound Uremic Toxins

Protein-bound uremic toxins also exert proinflammatory activities and can lead to vascular damage via activating the crosstalk between circulating leucocytes and vessels [50]. Indoles are protein-bound uremic toxins and are active metabolites derived from dietary tryptophan metabolism via gut microbiota. Indole-3 acetic acid (IAA), a tryptophan-derived uremic toxin, is negatively correlated with endothelial progenitor cell numbers in hemodialysis patients [49]. Indoxyl sulfate (IS) is one of the most well-known gut-derived uremic toxins and is significantly associated with mortality, congestive heart failure, vascular calcification, and dialysis vascular access thrombosis [51,52]. Another important protein-bound uremic toxin is para-cresyl sulfate (PCS), which is also derived from the metabolism of dietary aromatic amino acids (tyrosine or phenylalanine) via the gut bacteria.

IS has begun to be linked to PVD over the recent decade. Lin et al. reported that IS and PCS were associated with PVD [35]. They also found that total PCS levels could predict hemodialysis access viability in addition to traditional risk factors. Serum IS levels could also associate with adverse events in PVD. Wu et al. reported that serum-free IS levels predicted postangioplasty thrombosis of hemodialysis grafts [38].

More and more studies have revealed the underlying pathophysiology of the impacts of protein-bound uremic toxins on the development or progression of PVD. Both IS and PCS significantly promote calcifications in the aorta, femoral artery, and carotid artery via activation of proinflammatory and procoagulant pathways, which are associated with impaired glucose metabolism [40]. In addition, IS can suppress hypoxia-inducible factor (HIF)-1α and its downstream interleukin (IL)-10/signal transducer and activator of transcription (STAT) 3/vascular endothelial growth factor (VEGF) signaling in human endothelial colony-forming cells [39]. This partially underlies how IS impairs the proangiogenic activity of endothelial progenitor cells in PVD. Moreover, Kuo et al. also found that IS can impair neovascularization in ischemic limbs, which was induced by an angiotensin receptor blocker [41]. Furthermore, elevated IS in CKD could lead to a procoagulant state [53]. Bertrand et al. reported that patients with CKD had higher tissue factor activity and concentration [37]. Circulating IS and IAA levels were positively correlated with levels of tissue factor. IA and IAA increase the expression of tissue factor and its thrombogenic activity in endothelial cells and peripheral blood mononuclear cells. Besides, IA and IAA prolong the half-life of tissue factor in primary human vascular smooth muscle cells [36]. Acute thrombosis usually results in morbidity or mortality in patients with PVD [54]. Other studies have reported that PCS is significantly associated with atherosclerosis in hemodialysis patients and can promote oxidative stress-induced osteogenesis and vascular calcifications [42,43].

Taken together, current evidence shows that phosphorous and protein-bound uremic toxins are the most deleterious offenders to vascular injury leading to the development and progression of PVD. We will discuss the prevention or management of PVD with regard to these uremic toxins in the next section.

## 3. Reducing Uremic Toxins for Treatment and Prevention of PVD

Traditional treatment for PVD in CKD patients includes reducing the risk factors via smoke cessation [55], exercise [56], antiplatelet therapy (such as aspirin or clopidogrel) [57], lipid-lowering agents [58], cilostazol [16], and revascularization [14]. However, there is still a large gap of evidence on the effects of these treatments between patients without and with CKD, because most of these studies excluded patients with advanced CKD or ESRD. Moreover, the accumulation of uremic toxins during the process of progressive kidney disease may cause or exaggerate PVD. In the previous section, we have illustrated how uremic toxins can damage vessels and contribute to an incident or progressive PVD. Thus, to remove or reduce circulating levels of uremic toxins is a reasonable strategy to prevent or ameliorate PVD in patients with CKD. Here, we have also reviewed the current evidence of uremic toxin-lowering treatments for PVD, which is focused on the setting of CKD (summarized in Table 2).

### 3.1. Small Water-Soluble Uremic Toxins

Uremic toxin-reducing therapy has been proposed for the treatment of patients with uremia for decades. It is widely accepted that the adequacy of dialysis can be generally defined by a urea-reduction ratio of >65% or single-pool Kt/V_urea_ of >1.20 in patients on chronic hemodialysis or weekly Kt/V_urea_ of >1.70 in patients on chronic peritoneal dialysis [66,67]. However, no study has shown a treatment specifically decreases circulating small water-soluble uremic toxins and ameliorates or prevents PVD with the exception of circulating phosphorus. Phosphate-lowering treatments have shown their effects on preventing the development or progression of PVD. Finch et al. reported that uremic rats on a low-phosphate diet for three months significantly normalized calcium phosphate production, lowered serum parathyroid hormone, and reduced vascular calcification [30]. In addition, animal studies have shown that phosphate lowering via non-calcium-containing phosphate binders, such as lanthanum carbonate and sevelamer, could attenuate hyperphosphatemia and prevent vascular calcification in the aorta and peripheral arteries [60,61]. Moreover, lanthanum carbonate and sevelamer also slow the progression of vascular calcification in patients on dialysis compared with calcium-containing phosphate binders [59,62].

Several methods have been proposed to reduce TMAO levels, including dietary intervention, prebiotics and probiotics, antibiotic intervention, or fecal transplantation [68,69]. Besides, 3,3-dimethyl-1-butanol, a structural analogue of choline, has been reported to inhibit microbial trimethylamine formation and attenuate circulating TMAO levels, resulting in ameliorated foam cell formation and atherosclerosis in apolipoprotein E knockout mice [70]. However, these TMAO-reducing interventions have not proved to improve or prevent the development of PVD in patients with CKD.

### 3.2. Middle Molecular Uremic Toxins

A middle molecular uremic toxin, β2-microglobulin, was reported to associate with PVD in a case-control study [33]. Circulating β_2_-microglobulin levels were inversely associated with eGFR and could predict cardiovascular events (including peripheral ischemia events and surgery for PVD) in patients with CKD [34]. However, whether the removal of β2-microglobulin prevents or mitigates the development of PVD remains largely unknown. Although previous research has implied a potential mechanistic role of β_2_-microglobulin in PVD, more studies are needed to clarify the effect of removing small water-soluble or middle molecular uremic toxins on the development or progression of PVD.

### 3.3. Protein-Bound Uremic Toxins

Protein-bound uremic toxin-reducing therapy has been demonstrated for its therapeutic potential in delaying the progression of CKD [71,72]. However, there was little knowledge about whether removing or reducing these protein-bound toxins could also be beneficial to vascular calcification and PVD in CKD before two decades ago. Growing evidence has shown the potential of treatments reducing protein-bound uremic toxins in PVD. Previous studies reported an oral charcoal adsorbent of uremic toxins, AST-120 (Kremezin^®^, Kureha Corporation, Tokyo, Japan) could have beneficial effects on arterial stiffness and vascular calcification in patients with CKD [63,64]. Hung et al. demonstrated that AST-120 can reduce plasma IS, which was exaggerated in mice with subtotal nephrectomy [39]. The neovascularization after unilateral hindlimb ischemia was impaired in subtotal nephrectomy mice fed with indole for 12 weeks. Under hypoxia, VEGF increased through HIF-1α/IL-10/STAT3 signaling. IS decreased VEGF through the HIF-1α/IL-10/STAT3 pathway, then suppressed proangiogenic endothelial progenitor cells and induced endothelial cell dysfunction. Moreover, AST-120 rescued decreased endothelial progenitor cell mobilization and improved impaired neovascularization after hindlimb ischemia in mice with subtotal nephrectomy. A recent study also confirmed the role of IS in the pathophysiology of PVD in CKD. Shih et al. reported that uremic binding therapy using AST-120 given before and after arteriovenous fistula creation could also ameliorate and prevent the neointimal formation of arteriovenous fistulas in mice [65]. This oral charcoal adsorbent reversed the higher expression of matrix metalloproteinase-2, matrix metalloproteinase-9, tumor necrosis factor-α, and transforming growth factor-β induced by CKD in neointima tissue. These studies imply the beneficial role of IS-binding therapy in treating or preventing PVD in CKD.

The oral adsorbent AST-120 administration could also lower serum or plasma PCS levels in CKD rats [73,74]. Since AST-120 can absorb other gut-derived protein-bound uremic toxins such as IS, the beneficial effects of AST-120 on the neointimal formation of arteriovenous fistulas and neovascularization might also be partly owing to the reduction of the deleterious impact of PCS in the abovementioned studies. In addition, sevelamer hydrochloride and renal replacement therapy have been proven to reduce circulating PCS concentrations in patients with CKD [75,76,77,78]. Sevelamer treatment did not deteriorate vascular calcification in patients with hemodialysis compared with those treated with calcium-based phosphate binders [79]. However, a randomized controlled trial failed to demonstrate the beneficial effect of sevelamer on arterial stiffness in 120 patients with nondiabetic CKD [76]. Hemodialysis has a much higher removal rate of PCS than peritoneal dialysis per se, though PCS is poorly removed by both modalities because >90% PCS is bound to plasma protein [77,78]. It is also not known which dialysis modality could prevent the development of progression of PVD as well as reduce circulating PCS level.

Taken together, the latest evidence shows that phosphorus and IS are the most promising therapeutic or preventive targets for PVD in CKD. We still need more clinical evidence, such as well-designed clinical trials or prospective studies to prove the beneficial effect of these strategies on PVD in patients with CKD.

## 4. Future Perspectives

Previous studies have shown that uremic toxins are deleterious to the kidney, cardiovascular system, brain, lungs, and gut [22]. In the field of PVD, we are becoming aware of the adverse effects of phosphorus and protein-bound uremic toxins and the benefits of removing these uremic toxins.

Phosphorus and calcium are key players in the management of CKD-mineral and bone disorder (CKD-MBD) [80]. Controlling phosphate levels toward the normal range, limiting the use of calcium-based phosphate binders, and avoiding hypercalcemia may reduce vascular calcification, cardiovascular events, and mortality [81,82,83,84]. Although dietary phosphate restriction and non-calcium-based phosphate-lowering treatment decreased vascular calcification, slowed the progression of aortic calcification, and reduced mortality in animal studies and randomized controlled trials (Table 2), no randomized controlled trials or prospective studies with a prespecified primary endpoint of an incident or progressive PVD were conducted to compare the effect of phosphate-lowering treatment in CKD or dialysis patients. Besides, calcitriol, vitamin D analogue, and calcimimetics are important for managing CKD-MBD, especially in patients on dialysis [80]. The antiproteinuric and anti-inflammatory effects of vitamin D have a promising role in the protection of endothelium and podocytes in diabetic nephropathy [85]. Growing evidence also suggests a protective role of vitamin D in renal tubular diseases caused by inflammation [86]. Lower vitamin D level has been reported to be an independent risk factor for PVD [87]. However, a randomized controlled pilot study failed to demonstrate the beneficial effects of oral high-dose vitamin D on endothelial dysfunction and arterial stiffness in non-CKD patients [88]. More high-quality evidence remains needed to support these interventions as a standard treatment for PVD in CKD patients.

Growing evidence has shed some light on the role of protein-bound uremic toxins, especially IS and PCS, in the pathophysiology of PVD and has proposed a novel treatment for PVD by removing these uremic toxins. First, AST-120 is one of the most well-known oral adsorbents, which could bind various uremic toxins in the gastrointestinal tract, especially gut-derived protein-bound uremic toxins and it may have the potential to slow the progression of CKD [72,89]. With regard to PVD, there is more in vitro and in vivo evidence suggesting the potentially beneficial effects of AST-120 for PVD in patients with CKD (Table 2). We look forward to clinical trials to reveal the effect of AST-120 on PVD outcomes in patients with CKD or undergoing dialysis. Second, the effectiveness of dialysis on the removal of protein-bound uremic toxins has been investigated in patients on maintenance hemodialysis. Former research has shown that current dialysis modalities do not ensure a satisfactory removal of protein-bound uremic toxins such as IS or PCS, even with the use of hemodiafiltration [90]. However, dialysis time extension or daily hemodialysis may enhance the removal of protein-bound uremic toxins and maintain lower levels of IS and p-cresol [91,92]. A recent pilot study also reported an alternative hemodialysis model to enhance the removal of protein-bound uremic toxins [93]. The authors found that infusion of ibuprofen, a binding competitor that competes with protein-bound uremic toxins for binding sites of serum albumin, into the prefilter (arterial) bloodline augmented the dialytic removal of protein-bound uremic toxins such as IS and PCS and reduced their serum levels in 18 patients on maintenance hemodialysis [93]. These promising results imply the potential benefits of modern dialysis modalities for PVD by means of the removal of uremic toxins contributing to the disease though further clinical trials or prospective studies are warranted to confirm their effects on an incident or progressive PVD in patients undergoing dialysis. Third, targeting the gut microbiota has been studied to revert the vicious cycle between CKD and gut dysbiosis [94]. Previous studies have shown promising effects of foods as medicine on reducing uremic toxins and inflammation in CKD [94]. Foods rich in prebiotic fibers or polyphenols can increase the production of beneficial short-chain fatty acids and reduce the production of uremic toxins through the gut microbiota [94,95]. Additionally, prebiotics and probiotics may reduce circulating IS and PCS in patients with CKD by the modulation of intestinal flora [96,97]. Since we currently do not have the knowledge, whether prebiotics and probiotics or foods rich in prebiotic fibers and polyphenols have a beneficial effect on preventing or attenuating PVD in the CKD population merits further investigation. Finally, a recent study found that renal proximal tubular cells can increase the active secretion of IS by upregulating organic anion transporter-1 in response to plasm IS levels sensed by these tubular cells [98]. There is no proven effective treatment for enhancing renal IS excretion in CKD patients. Therefore, preserving renal function is important to maintain tubular secretion of IS and may result in less accumulation of the uremic toxin.

## 5. Conclusions

PVD is more prevalent in CKD than in the general population. PVD in patients with CKD is linked to poor prognosis and higher mortality. In addition to traditional risk factors of atherosclerosis (e.g., smoking, aging, hypertension, diabetes, and hyperlipidemia, etc.), CKD patients have nontraditional risk factors for PVD, such as uremic milieu, hyperphosphatemia, hyperparathyroidism, and oxidative stress. Studies have shown that uremic toxins, especially phosphorus and protein-bound uremic toxins, take part in the pathogenesis of PVD in CKD patients. Several pieces of evidence also support these uremic toxins as therapeutic targets for PVD. Phosphate restriction in diet or non-calcium-based phosphate binders attenuate vascular calcification. Oral charcoal adsorbent AST-120 removes gut microbiome-derived IS and PCS and attenuates vascular calcification, arterial stiffness, restored neovascularization, and halts neointima formation of arteriovenous fistulas. However, high-quality interventional studies (randomized controlled trials) are needed to determine the effect of these treatments on preventing or relieving PVD and to establish a new standard of care for PVD.

## Figures and Tables

**Table 1 toxins-12-00808-t001:** The role of uremic toxins in peripheral vascular disease (PVD) in experimental models or patients with chronic kidney disease (CKD).

Uremic Toxins	Authors (Publishing Year)	Subjects	Results	References
Small Water-Soluble				
Phosphorus				
	Boaz et al. (2005)	HD patients	Serum phosphorus independently predicts the development of PVD.	[26]
	Son et al. (2006)	Human aortic VSMCs	High inorganic phosphate induces calcification and apoptosis in VSMCs.	[27]
	Mozar et al. (2008)	Human PBMCs and RAW264.7 macrophages	High extracellular inorganic phosphate downregulates RANK-RANKL signaling and inhibits osteoclast differentiation.	[28]
	Ix et al. (2009)	Free of clinical apparent CVD regardless of CKD	Higher phosphorus levels are strongly associated with higher ABI values.	[29]
	Finch et al. (2013)	CKD rats	A high-phosphate diet increases aortic calcium and calcification in CKD rats.	[30]
	Jimbo et al. (2014)	CKD rats	Phosphate synergizes with FGF 23 to promote calcification in aorta and VSMCs.	[31]
TMAO	Matsumoto et al. (2020)	Rats	TMAO impairs relaxation of femoral arteries.	[32]
Middle Molecules				
β_2_-Microglobulin	Wilson et al. (2007)	PVD patients and controls	Plasma β_2_-microglobulin levels correlate with ankle–brachial index.	[33]
	Liabeuf et al. (2012)	CKD patients (stages 2 to 5D) and controls	Plasma β_2_-microglobulin levels are associated with cardiovascular events (MACE plus peripheral ischemia and surgery for PVD).	[34]
Protein-Bound				
IS	Lin et al. (2012)	HD patients	Serum IS level is associated with PVD.	[35]
	Chitalia et al. (2013)	Primary human VSMCs	IS increases tissue factor expression and half-life resulting in greater clot formation by inhibition of ubiquitination.	[36]
	Gondouin et al. (2013)	CKD patients (stages 3 to 5D)	Plasma IS levels are positively correlated with tissue factor levels.	[37]
		HUVECs and PBMCs	IS increases tissue factor expression and production. IS also enhances procoagulant activity of tissue factor.	
	Wu et al. (2016)	HD patients	Serum IS associates with dialysis graft thrombosis.	[38]
	Hung et al. (2016)	CKD mice	IS impairs endothelial progenitor cell function and inhibits neovascularization.	[39]
	Opdebeeck et al. (2019)	CKD rats	IS promotes calcification in the aorta and peripheral arteries.	[40]
	Kuo et al. (2020)	CKD mice	IS attenuates valsartan-induced neovascularization.	[41]
PCS				
	Lin et al. (2012)	HD patients	Serum level of PCS is associated with PVD.	[35]
	Jing et al. (2016)	HD patients and ApoE^-/-^ CKD mice	Elevated serum PCS levels are associated with carotid atherosclerosis. PCS promotes atherogenesis via increasing ROS.	[42]
	Opdebeeck et al. (2019)	CKD rats	PCS promotes calcification in the aorta and peripheral arteries.	[40]
	Chang et al. (2020)	HASMCs	PCS induces osteogenesis and uremic vascular calcification.	[43]

Abbreviations: ABI, ankle–brachial index; ApoE, Apolipoprotein E; CKD, chronic kidney disease; CVD, cardiovascular disease; FGF 23, fibroblast growth factor 23; HASMC, human arterial smooth muscle cell; HD, hemodialysis; HUVEC, human umbilical vein endothelial cell; IS, indoxyl sulfate; MACE, major adverse cardiovascular events; PCS, p-cresyl sulfate; PBMC, peripheral blood monocytic cell; PVD, peripheral vascular disease; RANK, receptor activator of nuclear factor *κ*B; RANKL, receptor activator of nuclear factor *κ*B ligand; ROS, reactive oxygen species; TMAO, trimethylamine N-oxide; VSMC, vascular smooth muscle cell.

**Table 2 toxins-12-00808-t002:** The effects of uremic toxin-targeting therapies on PVD in patients with CKD or in animal studies.

Uremic Toxins	Authors (Publishing Year)	Subjects	Interventions	Results	References
Phosphorus					
	Chertow et al. (2002)	HD patients	Sevelamer vs. calcium-based phosphate binders	Sevelamer treatment is linked to less hypercalcemia and less progression of aortic calcification.	[59]
	Neven et al. (2009)	CKD rats	Lanthanum carbonate	2% lanthanum carbonate reduces medial calcification in the aorta, carotid artery, and femoral artery.	[60]
	Finch et al. (2013)	CKD rats	Low-phosphate diet	Phosphate restriction attenuates aortic calcification and mortality.	[30]
	De Shutter et al. (2013)	CKD rats	Calcium carbonate/magnesium carbonate (CaMg) vs. sevelamer carbonate	Either CaMg or sevelamer carbonate controls hyperphosphatemia and prevents the development of aortic calcification.	[61]
	Wada et al. (2014)	HD patients	Lanthanum carbonate vs. calcium carbonate	Lanthanum carbonate attenuates the progression of vascular calcification.	[62]
IS					
	Nakamura et al. (2004)	Nondiabetic CKD patients	AST-120 (Kremezin) vs. none	Arterial stiffness (pulse-wave velocity) significantly decreases in the AST-120 group at 2 years.	[63]
	Goto et al. (2013)	Patients with stage 4-5 CKD	AST-120 vs. none	The aortic calcification index was significantly lower in patients with a 6-month AST-120 treatment.	[64]
	Hung et al. (2016)	CKD mice	AST-120	AST-120 lowers plasma IS and reverses the decreased endothelial progenitor cell mobilization and the impaired neovascularization.	[39]
	Shih et al. (2020)	CKD mice	AST-120	AST-120 decreases serum IS and prevents neointima formation of arteriovenous fistulas.	[65]

Abbreviations: IS, indoxyl sulfate; CKD, chronic kidney disease; PVD, peripheral vascular disease, PTH, parathyroid hormone.

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
