# Peer review of "Targeting Uremic Toxins to Prevent Peripheral Vascular Complications in Chronic Kidney Disease"

_toxins, 2020, doi:10.3390/toxins12120808_

Round 1
Reviewer 1 Report
Overall, this is a very clear and well-written review of the literature of uremic toxins and their contributing role on PVD in CKD.
Minor comments
In Tables 1 and 2, under results section, please put either a period or semi-colon at the end of each statement since it was hard to differentiate between lines.
Author Response
We would like to thank the reviewer for the comprehensive assessments, constructive criticisms, and valuable comments on our manuscript. Under your recommendations, we have revised our manuscript in response to all comments of the reviewers. We believe that our manuscript did gain more precision and clearness.
Overall, this is a very clear and well-written review of the literature of uremic toxins and their contributing role on PVD in CKD.
Minor comments
In Tables 1 and 2, under results section, please put either a period or semi-colon at the end of each statement since it was hard to differentiate between lines.
Response:
We thank the reviewer very much for the reminder. We have added a period at the end of each statement under the “Results” column in Tables 1 and 2.
Reviewer 2 Report
This interesting review about the role of uremic toxins to prevent peripheral vascular diseases in chronic kidney disease patient is well written and can give a potential relevant contribution to the current literature.
I suggest you few modifications to improve your work:
You should combine the paragraph 2.2 on the "middle molecular uremic toxins" to the 3.2 and add further informations on beta-2 microglobulin, I suggest you the paper "Plasma beta-2 microglobulin is associated with cardiovascular disease in uremic patients. Kidney Int. 2012 Dec;82(12):1297-303. doi: 10.1038/ki.2012.301. Epub 2012 Aug 15. PMID: 22895515".
In the "future perspectives paragraph", at line 242 when you write about the lower vitamin D status as an independent risk factor for PVD, I suggest you the paragraph 3.4 "Role of Vitamin D in Endothelium and Podocyte Preservation in DN" of the article "Role of Vitamin D Status in Diabetic Patients with Renal Disease. Medicina (Kaunas). 2019 Jun 13;55(6):273. doi: 10.3390/medicina55060273. PMID: 31200589; PMCID: PMC6630278." and the tubular protection as reported in the article
"Protective Role of Vitamin D in Renal Tubulopathies. Metabolites. 2020 Mar 19;10(3):115. doi: 10.3390/metabo10030115. PMID: 32204545; PMCID: PMC7142711".
Author Response
We would like to thank the reviewer for the comprehensive assessments, constructive criticisms, and valuable comments on our manuscript. Under your recommendations, we have revised our manuscript in response to all comments of the reviewers. We believe that our manuscript did gain more precision and clearness.
This interesting review about the role of uremic toxins to prevent peripheral vascular diseases in chronic kidney disease patient is well written and can give a potential relevant contribution to the current literature.
I suggest you few modifications to improve your work:
You should combine the paragraph 2.2 on the "middle molecular uremic toxins" to the 3.2 and add further informations on beta-2 microglobulin, I suggest you the paper "Plasma beta-2 microglobulin is associated with cardiovascular disease in uremic patients. Kidney Int. 2012 Dec;82(12):1297-303. doi: 10.1038/ki.2012.301. Epub 2012 Aug 15. PMID: 22895515".
Response:
We thank the reviewer for the excellent suggestion. We have added the information on b2-microglobulin based on the suggested paper in paragraph 2.2 and summarized the relevant part of paragraph 2.2 to 3.2. This information was also added in Table 1 (Page 3).
Lines 106 to 110:
This group further demonstrated the inverse-relationship between b2-microglobulin level and estimated glomerular filtration rate (eGFR) in patients with CKD [39]. In this study, b2-microglobulin levels were independently associated with cardiovascular events which were a composition of traditional major cardiovascular events plus peripheral ischemia and surgical procedures for PVD in pre-dialytic CKD patients [39].
Lines 190 to 196:
…Circulating b2-microglobulin levels were inversely associated with eGFR and could predict cardiovascular events (including peripheral ischemia events and surgery for PVD) in patients with CKD [34]. However, whether the removal of b2-microglobulin prevents or mitigates the development of PVD remains largely unknown. Although previous research has implied a potential mechanistic role of b2-microglobulin in PVD, more studies are needed to clarify the effect of removing small water-soluble or middle molecular uremic toxins on the development or progression of PVD.
In the "future perspectives paragraph", at line 242 when you write about the lower vitamin D status as an independent risk factor for PVD, I suggest you the paragraph 3.4 "Role of Vitamin D in Endothelium and Podocyte Preservation in DN" of the article "Role of Vitamin D Status in Diabetic Patients with Renal Disease. Medicina (Kaunas). 2019 Jun 13;55(6):273. doi: 10.3390/medicina55060273. PMID: 31200589; PMCID: PMC6630278." and the tubular protection as reported in the article "Protective Role of Vitamin D in Renal Tubulopathies. Metabolites. 2020 Mar 19;10(3):115. doi: 10.3390/metabo10030115. PMID: 32204545; PMCID: PMC7142711".
Response:
We agree with the reviewer and thank the reviewer for the suggestions. These two papers indeed provide additional valuable information to this review. However, the third section (paragraphs 3.1 to 3.3) is about therapeutics reducing uremic toxins to prevent PVD. We believe that it would be better that they stay in the “Future Perspectives”. We also added the two papers in this section.
Lines 250-252:
The antiproteinuric and anti-inflammatory effects of vitamin D have a promising role in the protection of endothelium and podocytes in diabetic nephropathy [85]. Growing evidence also suggests a protective role of vitamin D in renal tubular diseases caused by inflammation [86].